# mCRP-Induced Focal Adhesion Kinase-Dependent Monocyte Aggregation and M1 Polarization, Which Was Partially Blocked by the C10M Inhibitor

**DOI:** 10.3390/ijms25063097

**Published:** 2024-03-07

**Authors:** Ylenia Pastorello, Doina Manu, Xenia Sawkulycz, Vittorio Caprio, Claudia Banescu, Minodora Dobreanu, Lawrence Potempa, Mario Di Napoli, Mark Slevin

**Affiliations:** 1Department of Anatomy and Embryology, George Emil Palade University of Medicine, Pharmacy, Science and Technology of Targu Mures, 540142 Targu Mures, Romania; ylenia.pastorello@gmail.com; 2Doctoral School of Medicine and Pharmacy, George Emil Palade University of Medicine, Pharmacy, Science and Technology of Targu Mures, 540142 Targu Mures, Romania; 3Center for Advanced Medical and Pharmaceutical Research (CCAMF), George Emil Palade University of Medicine, Pharmacy, Science and Technology of Targu Mures, 540142 Targu Mures, Romania; doina.manu@umfst.ro (D.M.); claudia.banescu@umfst.ro (C.B.); minodora.dobreanu@umfst.ro (M.D.); 4Department of Life Sciences, Manchester Metropolitan University, Manchester M15 6BH, UK; xenia.sawkulycz@stu.mmu.ac.uk (X.S.); v.caprio@mmu.ac.uk (V.C.); 5Department of Life Sciences, College of Science, Health and Pharmacy, Roosevelt University, Schaumburg, IL 60173, USA; lpotempa01@roosevelt.edu; 6Neurological Service, dell’ Annunziata Hospital, Sulmona, 67039 L’Aquila, Italy; mario9dinapoli@katamail.com

**Keywords:** macrophage M1 polarization, mCRP, inflammation, vascular activation, small molecule inhibitors

## Abstract

Monomeric C-reactive protein (mCRP) has recently been implicated in the abnormal vascular activation associated with development of atherosclerosis, but it may act more specifically through mechanisms perpetuating damaged vessel inflammation and subsequent aggregation and internalization of resident macrophages. Whilst the direct effects of mCRP on endothelial cells have been characterized, the interaction with blood monocytes has, to our knowledge, not been fully defined. Here we showed that mCRP caused a strong aggregation of both U937 cell line and primary peripheral blood monocytes (PBMs) obtained from healthy donors. Moreover, this increase in clustering was dependent on focal adhesion kinase (FAK) activation (blocked by a specific inhibitor), as was the concomitant adhesive attachment to the plate, which was suggestive of macrophage differentiation. Confocal microscopy confirmed the increased expression and nuclear localization of p-FAK, and cell surface marker expression associated with M1 macrophage polarization (CD11b, CD14, and CD80, as well as iNOS) in the presence of mCRP. Inclusion of a specific CRP dissociation/mCRP inhibitor (C10M) effectively inhibited PBMs clustering, as well as abrogating p-FAK expression, and partially reduced the expression of markers associated with M1 macrophage differentiation. mCRP also increased the secretion of pro-inflammatory cytokines Interleukin-8 (IL-8) and Interleukin-1β (IL-1β), without notably affecting MAP kinase signaling pathways; inclusion of C10M did not perturb or modify these effects. In conclusion, mCRP modulates PBMs through a mechanism that involves FAK and results in cell clustering and adhesion concomitant with changes consistent with M1 phenotypical polarization. C10M has potential therapeutic utility in blocking the primary interaction of mCRP with the cells—for example, by protecting against monocyte accumulation and residence at damaged vessels that may be predisposed to plaque development and atherosclerosis.

## 1. Introduction

Inflammation contributes to the development of atherosclerosis, manifesting most commonly in unstable (haemorrhagic, heterogeneous vascular, and inflammatory) plaque development and acute thrombosis. Whilst acute inflammatory reaction is strongly associated with plaque rupture and myocardial infarction (MI), chronic inflammation, such as that seen in autoimmune disease, is linked to earlier onset or acceleration of existing cardiovascular disease (CVD) [1]. The earliest stages of damage involve the shear stress-mediated creation of endothelial cell (EC) blood vessel intimal ‘hot spots’ of activation, where the initial damage results in focalised inflammation, the attraction of activated platelets, and monocyte macrophages, concomitant with their polarization to the M1 phenotype [2,3].

Recently, native C-reactive protein (nCRP), and, specifically the dissociated monomeric form (mCRP), has been inextricably, and significantly, linked to more rapid advancement and worse predictive outcomes at all stages of disease [4]. Initiation and perpetuation of the inflammatory milieu are directly correlated with rising levels of CRP. The hepatic production of this pentameric compound relies on augmented interleukin-6 (IL-6) circulating degrees to regulate CRP transcription during infectious and inflammatory states [5]. Bioactive properties of CRP are mainly ascribed to its dissociated, monomeric configuration; conversion to mCRP takes place as a consequence of the interaction with exposed phosphocholine groups present on activated cell membranes [6].

A direct association between increased plasma levels of mCRP (but not serum levels of high-sensitivity CRP and IL-6) and carotid plaque number and height was described by Melnikov et al. during a seven-year follow-up [7]. mCRP promotes pro-inflammatory macrophage phenotype development, as well as the upregulation and expression of tumor necrosis factor alpha (TNF-α) and interleukin-1β (IL-1β). Localization of mCRP is displayed during ischemic stroke and MI leading, in the latter scenario, to impairment of myocardial repair by prolonging M1 macrophage residue in the affected tissue [8].

Balanced activation of the signal transducers and activators of transcription (STAT1 and STAT3/STAT6) plays a pivotal role in regulation of macrophage polarization. M1 pro-inflammatory phenotype pathways are governed by activation of nuclear factor kappa B (NFĸB) and STAT1, with the former representing the main transcription factor in modulating the expression of inflammatory genes, such as TNF-α, IL-1β, cyclooxygenase 2 (COX2), and IL-6 [9]. In addition, macrophage behavior is influenced by focal adhesion kinase (FAK), the cytoplasmic tyrosine kinase which, by functioning as a signaling scaffold, recruits monocytes/macrophages to the inflammation site. In this regard, in vivo myeloid-specific conditional FAK knockout murine models displayed impaired chemotaxis of CD11b-positive monocytes/macrophages (M1), compared to the control group [10]. Harada et al. (2017) showed that FAK-induced M1 macrophage secretion of matrix metallopeptidase 9 (MMP-9) and monocyte chemoattractant protein-1 (MCP-1) is, in a NFĸB-dependent fashion, concomitant with increased polarized motility [11]. 

In this article, we show FAK-dependent mCRP modulation of monocyte attachment and clustering is concomitant with differentiation into M1 macrophage sub-populations. This process was partly abrogated by the mCRP inhibitor C10M. However, the inhibitor did not block mCRP-induced secretion of pro-inflammatory cytokines interleukin-8 (IL-8) or interleukin 1-β (IL-1β).

## 2. Results

### 2.1. mCRP-Induced Monocyte Aggregation, Which Was Concomitant with Increased Expression of FAK and M1 Phenotype Transition

Our preliminary results show that mCRP (100 µg/mL) induced significant U937 aggregation at between 3 h and 24 h (Figure 1A–I; all images taken at ×100 and size bars included). Neither controls (A–B), nCRP (C–D) nor LPS (G–H) produced any noticeable clustering of the cells. Cluster size was largest in monocytes treated with mCRP for 24 h (F-arrow), although clusters were evident even after 3 h treatment (E-arrow). Mean aggregation size of clusters is shown in (I). The images in Figure 1 and Figure 2 appear slightly blurred because normal brightfield microscopy was used to image the cells over the focal plane of the medium (floating and attached to the plastic).

Continuing the investigation with a primary source of freshly isolated human peripheral macrophages, we confirmed a similar pattern of mCRP-induced aggregation after 24 h, as shown in Figure 2. Here we incorporated the CRP dissociation inhibitor-C10M into the study to observe any inhibitory capacity in relation to monocyte clustering. Figure 2A shows control cultured peripheral monocytes (all images taken at ×100 with size bars included) with no clustering that have a regular shape and size. Figure 2B shows the introduction of C10M at 100 µg/mL with 2 h pre-incubation, as used in previous studies, had no effect on the clustering or appearance of the cells. Following the addition of mCRP, also at 100 µg/mL (24 h), (Figure 2C) it is noticeable there are floating clusters of cells (red arrows), as well as larger, morphologically distinct and adherent (to the base of the culture plates) clumps (blue arrows) in the representative sample (Volunteer A).

Following pre-incubation with C10M for 2 h, the floating clusters were no longer visible, although small clumps of adherent cells remained (Figure 2D; blue arrows). Since FAK is a key protein orchestrating cell movement and attachment, we examined the effect of the FAK inhibitor Y397 on the aggregation process. Figure 2E shows no clustering in the presence of the inhibitor used at 10 µg/mL (24 h), whilst Figure 2F (2 h pre-incubation) shows complete inhibition of clustering and adherence induced by mCRP (100 µg/mL, 24 h exposure). Figure 2A–F were taken at ×100 magnification. Note that the FAK inhibitor was partially effective at 1 µg/mL, optimised inhibition occurred at 10 µg/mL, but the monocytes showed signs of loss of viability at 100 µg/mL, with the result that 10 µg/mL was the chosen inhibitor concentration. Since images were taken with light microscopy, the depth of the field was limited and hence the focus between adherent and floating cells is imperfect. The experiment was repeated twice, and a representative example is shown.

Topartially elucidate the molecular mechanisms responsible for inducing aggregation and/or changes in motility, we performed FACS analysis and confocal microscopy. Human peripheral monocytes treated with mCRP (100 µg/mL; 24 h) and photographed at ×100 magnification expressed significantly more p-FAK, as shown by the increased intensity of staining per cell and increased number of p-FAK positive cells (Figure 3C,D), when compared with control cells (Figure 3A) or those treated with C10M (Figure 3B). In Figure 3D a magnified image of mCRP-treated cells is shown, in which the change in morphological appearance and increased extent of FAK expression are very apparent (Figure 3C,D; ×100 and 200 respectively). Pre-incubation for 24 h with the CRP inhibitor C10M (100 µg/mL) partially abrogated the mCRP-induced p-FAK expression and nuclear translocation (Figure 3E). The inclusion of the FAK inhibitor alone had no effect on control expression, but did however totally block the increased expression of FAK induced by mCRP (Figure 3F,G, respectively). The mean intensity of staining per field of view is shown in Figure 3H and is representative of three measured areas from one well. This showed significance for the increase in intensity of FAK induced by mCRP (*p* ≤ 0.05 *), and both C10M (100 µg/mL) and the FAK inhibitor (10 µg/mL) blocked the effect (*p* ≤ 0.01 **).

Cell surface markers of the monocyte-macrophage phenotype were assessed by FACS. Exposure of monocytes, cultured in SFM, to mCRP (100 µg/mL; 24 h) resulted in increased cell surface expression of CD11b, CD14 and CD80 (15–27%; 31–63%; and 25–69% respectively), indicating a notable M1 phenotypic shift (Figure 4A,B). In the presence of the C10M inhibitor (100 µg/mL), only CD80 showed a 50% reduction in cell surface expression, compared to treatment with mCRP alone (Figure 4C). In addition, iNOS expression was increased in mCRP-treated human peripheral monocytes (Figure 4B). Pre-incubation of PBM with the C10M inhibitor abrogated the M1-associated transition induced by mCRP (70% reduced back down to 46%). Almost a quarter (23%) of the control, untreated, cells expressed iNOS (Figure 4C). Results are shown from a representative experiment of 2 carried out using PMBs obtained from Volunteer A.

### 2.2. Western Blotting Showed That mCRP Reduced MAP-Kinase Phosphorylation but Increased p53 after 8 Min Exposure

The inflammatory activity of monocytes/macrophages, and their polarization, is often preceded by changes in cell signal transduction activation, particularly within the MAP-kinase cascade involving phosphorylation of ERK1/2, JNK, and/or p38. Here, human peripheral-derived monocytes were differentiated into adherent macrophages in the presence of PMA (50 µg/mL/72 h), then cultured in SPM for 48 h, and were then treated with mCRP for 10 min, with or without C10M inhibitor (100 µg/mL; 2 h pre-incubation). A reduction in pERK-1, and pJNK-1/2 levels, compared to the housekeeping control, (GAPDH) was observed (Figure 5A). Co-incubation of cells with C10M did not notably alter the protein phosphorylation levels. Interestingly, incubation with mCRP increased the expression of the pro-apoptotic marker p-p53 (1.7-fold), and this increased further when C10M was present in the medium (2.6-fold). Figure 5B shows image intensities obtained by Image Lab software.

### 2.3. ELISA Indicated That mCRP-Treated Macrophages Increased the Expression of IL-8 and IL-1β

Secretion of the pro-inflammatory cytokines TNF-α, IL-8, and IL-1β was measured in the SFM of human peripheral monocyte-differentiated (PMA) macrophages, after exposure to mCRP (100 µg/mL) ± C10M (100 µg/mL for 24 h. Inclusion of mCRP resulted in a notable increase in the presence of only IL-8 and IL-1β, as determined by ELISA (Figure 6). Pre-incubation with C10M had no effect on mCRP-induced IL-1β or IL-8. Analysis was conducted in triplicate wells (as defined by R&D systems protocol)—a representative example of the results from two similar experiments is shown.

## 3. Discussion

Over the last few years, mCRP has been shown to directly influence the process of inflammation, specifically promoting a hyperinflammatory status associated with heightened immune response to injury or disease. Sproston and Ashworth (2018) eloquently summarized the role of mCRP in mediating vascular-based recruitment and activation of leukocytes by stimulating nitric oxide (NO) and various cytokines, including IL-6, IL-1β, and TNF-α [12]. The critical role of monocytes/macrophages in controlling the immune response to insult necessitates a more detailed understanding of the dynamics and interactions with mCRP. Melnikov et al. summarized current evidence supporting a function of mCRP in monocyte recruitment, potential macrophage M1 polarization, and associated endothelial cell hyperactivation and dysfunction [4,13]. These aberrant mechanisms will most likely contribute significantly to more rapid development of life-threatening conditions, particularly atherosclerosis, unstable atherosclerotic plaques, and increased risk of thrombosis [7]. 

Here, for the first time, we showed that mCRP, but not nCRP nor LPS, caused monocyte aggregation in vitro, over a period of 3–24 h, with increasing cluster size over time. This occurred concomitant with elevated p-FAK expression and perinuclear to nuclear translocation, which was completely abrogated in the presence of the C10M inhibitor. Aggregation of monocytes/macrophages is surprisingly unaccounted for within the literature, even though their clustering and combination with platelets notably enhances vascular luminal damage and thrombotic tendency. Vincent et al. [14] demonstrated that macrophage aggregation occurred in response to streptococcal infection via leukotriene B4 in a NFĸB-dependent fashion, whilst Biros et al. associated Interferon-γ (IFN-γ)-induced macrophage aggregation with increased inflammatory activity and multinucleation [15]. Interestingly, Locke et al. found that human monocyte cellular aggregates were produced in the presence of Interleukin-13 (IL-13) at the same time as M1 to M2 polarization [16]. 

In this study, mCRP-induced monocyte aggregation was associated with M1 polarization of human peripheral blood monocytes, and increased secretion of pro-inflammatory cytokines IL-8, and IL-1β. Therefore, there may be several interrelated signaling cascades responsible for mediating this process that have alternative outcomes. FAK is a critical regulator of cell adhesion and motility, interacting with cell surface integrins and adhesion molecules to modulate movement and adherence. Following nuclear relocalization, FAK acts as a scaffolding protein regulating transcription of early response genes, and interacts with cell survival modifying proteins, such as p53, where it protects against cell death [17]. Therefore, the nuclear translocation seen here that was induced by mCRP could be associated with increased survival time of monocytes in vivo, resulting in a more severe acute or chronic inflammatory response to injury. Cell counts indicated a notably high number of U937 cells in the mCRP-treated samples, compared to control. 

In this study we showed that pre-incubation with the FAK inhibitor I (CAS 4506-66-5 VWR) for 24 h prior to treatment with mCRP abrogated the U937 aggregation, indicating a FAK-dependent mechanism. Previous studies have demonstrated that CRP binding through α2β1 integrins activated FAK, and subsequently MAP kinase pathways, resulting in increased breast cancer cells adhesion and invasion [18]. In macrophages, FAK modulation is known to be critical for initiation and perpetuation of adhesion and motility, and although interactions through integrin activation predispose macrophage to M1 phenotypical polarization, our study is the first to show a direct link between mCRP, FAK, and monocyte aggregation, and subsequent polarization to the pro-inflammatory phenotype [19,20]. Zmuda and Pathak recently evidenced that only M1 polarized macrophages cultured in 3D hydrogels could induce signaling cues that promote epithelial/macrophage clustering, and it could therefore be the case that mCRP-induced cell-to-cell cohesion could be implicated in abnormal pathological responses within damaged tissue [21]. 

Zeller et al. provided evidence showing that their C10M inhibitor effectively abrogated both pCRP and mCRP-induced human peripheral blood monocyte adhesion, pro-inflammatory cytokines production, and endothelial macrophage interaction. However, in our work C10M, used under the same experimental conditions, reduced iNOS (responsible for monocyte–leukocyte adhesion, generation of reactive oxygen species, and enhancement of the pathological microenvironment), and reduced the mean cellular FAK expression, whilst blocking nuclear localization. C10M did not inhibit mCRP-induced monocyte M1 surface markers, including CD14 and CD80, and nor did it block the polarization of subsequently PMA-differentiated macrophages to M1, as demonstrated by a lack in reduction of the expression of pro-inflammatory cytokines IL-1β and IL-8 [22,23]. 

In other cells, for example EC, nuclear translocation of FAK results in degradation of transcription factors that promote pro-inflammatory cytokine translation; however in monocytes/macrophages nuclear translocation of FAK has been shown to induce phosphorylation of NFĸβ, in association with increased IL-6 expression, promoting an inflammatory phenotype and pseudopodic/phagocytic capability [24,25,26]. 

mCRP altered MAP kinase signaling of our quiescent, while PMA differentiated monocytes, as shown by Western blotting. Here, an unexpected reduction in phosphorylation of MAP kinase pathways ERK-2 (ERK-1 was unchanged) and, JNK 1/2 was seen, whilst the C10M inhibitor did not further modulate these cells signaling pathways. Recently, He et al. demonstrated in a detailed quantitative proteomics study that an increase in MAP kinase (c-Raf-MEK) signaling, in the form of a ‘spike’, was necessary for M2 macrophage polarization but was not required for changes towards the M1 phenotype. This is fitting with our data, where in fact a reduction in phosphorylative activity was seen in the presence of mCRP whilst increasing the prevalence of M1 pro-inflammatory phenotype. In this case, an acute increase in ERK1/2 would therefore have been associated with a shift to the anti-inflammatory phenotype, which was not seen in our cells [27]. Further studies should be carried out to look at the time course of changes from acute (minutes) to chronic (hours; where a later increase in p-ERK1/2 might be suggestive of pro-inflammatory activation), which may provide the missing information about monocyte to macrophage phenotype and differentiation. Regarding p-JNK1/2, Zha et al. [8], showed a moderate increase of phosphorylation of the protein by using the leukemia cell line THP-1, but they only examined the effects after chronic exposure to mCRP (24 h); to the best of our knowledge, this is the first study to show effects after acute exposure in primary peripheral blood monocytes that agrees with the generally observed effect of a reduction in MAP kinase activity. It should be investigated by further studies.

Phosphorylation of p53 however increased in the presence of mCRP. Whilst CRP (not mCRP), has been previously linked to activation of pro-apoptotic signaling pathways in macrophages from atheromatous plaques, to our knowledge, mCRP-associated apoptotic signaling in macrophages has not been studied [28]. Our results indicate a potential pro-apoptotic role, but this needs to be confirmed by further studies. In addition, C10M exacerbated phospho-p-53 expression, which would need to be considered if this SMI was to be considered potentially suitable for therapeutic use. 

C10M was not able to reduce the pro-inflammatory effect of mCRP on differentiated adherent macrophages, as demonstrated by ELISA (IL-1β and IL-8). Of note, however, IL-1β was increased in the supernatant of macrophages that were only treated by C10M, 24 h after initiation of the experiment. The effects of C10M were not investigated prior to this study, although Zeller et al., in using an indirect activation consisting of ADP-stimulated platelets, showed that C10M reduced IL-1 beta expression in mCRP-induced peripheral blood-derived monocytes, when measured by flow cytometry and confocal microscopy [22]. In our system, we can only hypothesize that the binding of C10M to the membrane of the monocytes stimulated the release of the active form of IL-1β from the intrinsic cytoplasmic store; however, further studies are needed to confirm this effect of C10M and its relevance, if any. Whilst LPS consistently significantly increased the secretion of IL-8 and IL-1β, a possible anomaly was observed whereby a reduction in TNF-α was observed after 24 h treatment within the same sample cohort. The only possible explanation would be that the relatively low concentration of LPS (10 ng/mL) we used in our experiments was insufficient to activate this cytokine response and pathway; however, this requires further investigation. TNF-α levels were also not increased in the presence of mCRP in our data sets, which in part agrees with the work of others, where neither TNF-α nor IL-6 notably increased in U937 monocytes exposed to mCRP for 24 h [29,30].

Our results suggest that this inhibitor may primarily act on monocytes rather than tissue-associated macrophages; however further work, which defines the mechanisms of mCRP-induced aggregation and impact in vivo, and draws on proof-of-concept data confirming [or not] the potential therapeutic utilization of C10M or C10M-like small molecule inhibitors, needs to be carried out to confirm these findings.

Since vascular damage is exacerbated through focalized inflammatory activity and intimal EC penetration signifies the early stages of atherosclerotic plaque formation, protecting the EC barrier from mCRP-associated build-up of immune cell or platelet aggregates should act as a possible therapeutic protection against large vessel stenosis and thrombosis. Understanding the mechanisms and capability of mCRP-induced immuno-aberrational activity should support the future development of novel anti-inflammatory treatments.

## 4. Materials and Methods

### 4.1. Production of mCRP

mCRP was provided by our collaborator Prof. Lawrence Potempa (Roosvelt University, Schaumburg, IL, USA) and was characterized, purified and tested for negative LPS, as described in Slevin et al. (2010) [31]. mCRP was used at concentrations of 100 µg/mL, based upon our previously published studies, preliminary data and the knowledge that acute traumatic inflammation, as seen in sepsis/autoimmune flareup, is associated with systemic circulating levels of CRP at this concentration [32].

### 4.2. Synthesis of the Inhibitor C10M

C10M was synthesized in the Department of Life Sciences, Manchester Metropolitan University (UK), according to the method described by Zeller et al. (2023), with modifications described below [22]. Preparation was achieved in a two-step sequence.

#### 4.2.1. Diethyl-(3-(dibutylamino)propyl) Phosphonate

Dibutylamine (1.6 g, 12.5 mmol) was added to a solution of diethyl (3-bromopropyl) phosphonate (7.1 g, 27.5 mmol) and sodium iodide (0.15 g, 0.75 mmol) in DMF (40 mL) at 0 °C. The mixture was stirred at 100 °C for 5 h then cooled to room temperature, and then acidified by addition of 1M aqueous HCl (50 mL). The aqueous phase was washed with ethyl acetate (2 × 40 mL) and then neutralized by the addition of solid sodium carbonate and extracted with ethyl acetate (2 × 50 mL). The combined organic phases were dried with anhydrous magnesium sulfate and concentrated. The residue was purified by column chromatography, using dichloromethane: methanol (9:1) as eluent to give the title compound as a colorless oil (1.0 g, 26%).

#### 4.2.2. (3-(dibutylamino)propyl) Phosphonic Acid (C10M)

Trimethylsilylbromide (10.0 g, 65.2 mmol) was added, dropwise, to a solution of diethyl-(3-(dibutylamino)propyl) phosphonate (1.0 g, 3.25 mmol) in dichloromethane (90 mL) at 0 °C. The mixture was stirred under reflux for 12 h, then cooled to room temperature. Water (150 mL) was added, and the aqueous phase washed with ethyl acetate (2 × 100 mL), and then concentrated under reduced pressure to give the title compound as a tan-coloured, viscous oil (0.60 g, 74%). ^1^H NMR data was in accordance with that published [32].

### 4.3. U937 Cell Culture and Differentiation

U937 cells, maintained in RPMI 1640, were supplemented with antibiotics (PSG) and 10% Fetal Bovine Serum (FBS) under culture at 37 °C, with 5% CO_2_. Prior to use, cells were counted in a Coulter. Monocytes were differentiated into macrophages using phorbol-12-myristate 13-acetate (PMA; 50 ng/mL; 72 h), after seeding in 6-well plates, at a concentration of 2 × 10^6^/mL (2 mL per well), before washing in Dulbecco’s Phosphate Buffered Saline (DPBS) and applying experimental growth conditions. For testing, cells were cultured in low serum-containing medium (2%) for 48 h prior to exposure to the mCRP inhibitor (C10M-100 µg/mL; 24 h), and/or mCRP (100 µg/mL, as used in our previous studies) [33]. 

The U937 cell line was utilised as preliminary confirmation of the ability of mCRP to induce aggregation of monocytes. Following this confirmation, further studies used primary human peripheral blood monocytes, because of being a more representative model of vascular and cardiovascular complications. Blood monocytes were extracted from 2 donors and used as described below.

### 4.4. Preparation and Culture of Human Peripheral Blood Monocytes

A total of 5 mL of anticoagulated blood was added to 5 mL of balanced salt solution and mixed by inversion. The blood sample was then layered onto 8 mL of Ficoll-Paque medium solution and centrifuged for 14 min at 400× *g*, followed by removal of the upper layer, consisting of platelets and plasma. The remaining monocytes were transferred into a tissue culture flask and cultured in RPMI 1640. PBMCs were cryopreserved with 10% dimethyl sulfoxide (DMSO) at −140 °C and thawed and cultured in RPMI-1640 medium containing L-glutamine supplemented with 1% penicillin/streptomycin and 10% fetal bovine serum at a density of 1–2 × 10^6^ cells/mL for experimentation. For testing, cells were cultured in low serum-containing medium (2%) for 24 h prior to exposure to the mCRP inhibitor (C10M—100 µg/mL; 24 h) and/or mCRP (100 µg/mL; 8 min for Western blotting, 2 h for FACS, and 24 h for ELISA and confocal microscopy). The FAK inhibitor-1 (Avantor/VWR; 1,2,4,5-benzenetetraamine, 4HCl; IC50 10 µM) was included in the aggregation and confocal microscopy investigations at concentrations used in previous studies. Use of HPBM was authorised with local UMFST, ethical approval pertaining to the project code (decizia comisie de etică a cercetării științifice nr. 2158 din 1 March 2023) was obtained, and the two volunteers were healthy, female, and aged 24-years-of-age (A) and 57-years-of-age (B), respectively. Representative figures are shown from monocytes obtained from A. Ethical approval for use of the volunteers as donors of peripheral whole blood was granted by the local university (UMFST) ethical committee, which approved the ongoing project “CRE-DICARD” (Financing contract nr. PCE 60/2021, project code: PN-III-P4-ID-PCE-2020-1540).

### 4.5. Monocyte Aggregation

U937 cells/primary peripheral blood monocytes or PBMs were cultured in serum-free medium for 24 h, prior to treatment with LPS (10 ng/mL), mCRP (100 µg/mL), or nCRP (100 µg/mL) for a further 24 h. Images were analyzed for the presence of aggregated clusters, and compared to control untreated cells at 3, 6, and 24 h. Significant differences in cluster size or number were determined using ImageJ Version 1.53 software, and SPSS Version 27 statistical analysis package (*p* ≤ 0.05 indicated a significant difference between groups; *p* ≤ 0.05 *; *p* < 0.01 **; *p* < 0.001 ***, *p* < 0.0001 ****, using ANOVA). Adhered clumps of cells that may represent differentiating cells were also noted. Three separate slides (biological replicates) and two regions per slide (×100; magnification) were used for the statistical analysis.

### 4.6. FACS

Human peripheral-derived monocytes, with or without mCRP (100 µg/mL; 24 h), ±C10M (100 µg/mL; 2 h pre-incubation) and with or without the FAK Y397 inhibitor (1–100 µg/mL, 2 h pre-incubation), were centrifuged and washed with RPMI-1640 medium. After washing in PBS, the cells were fixed in 2% paraformaldehyde (30 min) and were simultaneously stained with cell surface antibodies recognizing CD14, CD80, iNOS, CD11b, (BD Bioscience, Franklin Lakes, NJ, USA), and SYTOX AADvanced (Invitrogen, Waltham, MA, USA). For flow cytometry acquisition at least 100,000 mononuclear cells were analysed for each blood sample, and monocytes were gated in FSC/SSC (gating points were standardised for each of the cell surface markers). From monocytes, the expression of CD14, CD80, CD11b and iNOS were acquired for analysis as monoparametric histograms. The staining procedure was carried out according to the manufacturer’s instructions. The experiment was repeated twice and a representative example is shown in the results referring to PBMs obtained from volunteer A. 

### 4.7. Confocal Microscopy

Cells were firstly rehydrated with PBST (0.05%) and any potential non-specific binding was blocked (4% goat serum; 30 min). After two 5-minwashes in PBS, the cells were incubated with p-FAK at 1:500 dilution (1 h). After two further washes in PBS, slides were incubated with goat anti-mouse secondary antibody (1:200; 30 min; Alexa Fluor 488) and, after a final wash in PBS, the slides were mounted in vector shield containing DAPI. Fluorescence was analyzed using a Z1 AxioObserver confocal microscope (Zeiss, Oberkochen, Germany). Three coverslips were analyzed for each condition (and three random areas per slide recorded at ×100 for statistical evaluation) and the experiment was repeated twice, with a representative example shown using PBMs obtained from volunteer A. Mean fluorescent intensity was measured using AxioObserver software (ZEN 2.3 SP1). Differences in fluorescent expression were analyzed using one-way ANOVA with Bonferroni post-test analysis (*p* ≤ 0.05 *; *p* < 0.01 **; *p* < 0.001 ***, *p* < 0.0001 ****, using ANOVA).

### 4.8. ELISA

Following the manufacturer’s protocol, IL-8, TNF-α, and IL-1β were quantified in the supernatant by ELISA kits (R&D Systems, Minneapolis, MN, USA) following mCRP (100 µg/mL) and/or C10M (100 µg/mL) incorporation into the PBMC medium for 24 h. Lipopolysaccharide (LPS) (10 ng/mL; 24 h) was used as a positive control for macrophage cytokine production. Samples were tested in triplicate and, as per the manufacturer instructions, and results were presented as the mean from an experiment (with three biological replicates)—a representative example of two independent experiments is shown in the results, using PBMs from volunteer A. 

### 4.9. Western Blotting

PMA-differentiated human peripheral macrophages (2 × 10^6^/mL; 2 mL per well), cultured in SFM for 48 h, were treated with CRP (100 µg/mL;10 min), with or without a 2 h pre-incubation with C10M-(100 µg/mL). Due to the short acute treatment times, we used GAPDH as a protein control, as both we and others found no difference in total specific protein expression within this time frame resulting from a lack of translation, and then lysed by scraping into 500 µL of ice-cold RIPA buffer. The BCA protein assay was used to equilibrate the protein concentration, and samples were frozen at −80 °C until ready for use. 30 μg of each sample was mixed with 2× Laemmli sample buffer, boiled for 15 min, and loaded onto 12% SDS-PAGE pre-prepared gels; after separation, proteins were electrotransferred onto PVDF. After blocking with bovine serum albumin (BSA) (1% in TBS—Tween; pH 7.4) for 1 h at room temperature, filters were stained with antibodies to p-p53, p-JNK, and p-ERK1/2 (1:250). 

The filters were then diluted in the blocking buffer and incubated overnight at 4 °C on a rotating shaker. Following washing in TBS—Tween (5 × 10 min), the filters were stained for 1 h in TBS—Tween with 5% milk protein containing goat anti-rabbit or mouse HRP-conjugated secondary antibodies, and protein was then visualized using Image Lab Version 5.2.1 software. 

Western blotting was repeated twice—the bar graphs of the representative example show semi-quantitative data and fold differences, compared with the control (=1.0) PBMs obtained from volunteer A. 

For all experiments referred to above, when statistical analysis was employed, at least 3 replicates were analysed using SPSS Version 27 software; the mean/median ± SEM calculated (*p* ≤ 0.05 * or *p* ≤ 0.01 ** *p* ≤ 0.001 *** and *p* ≤ 0.0001 ****, which were designated as significant by ANOVA.

## Figures and Tables

**Figure 1 ijms-25-03097-f001:**
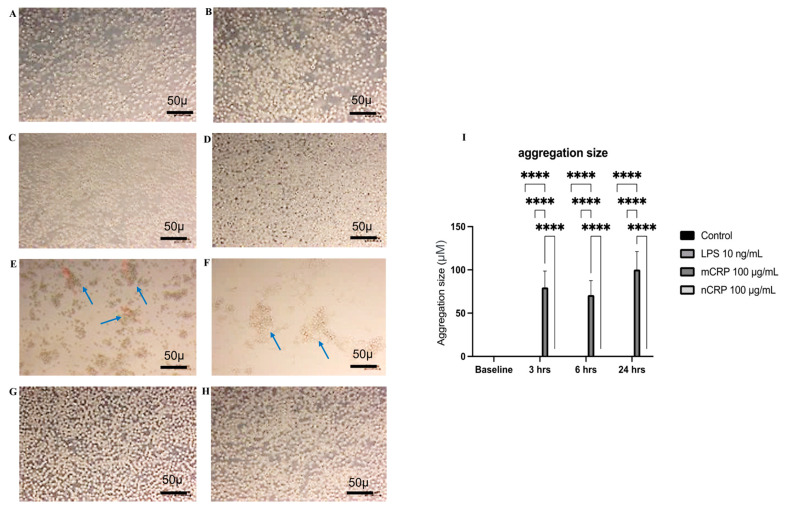
U937 monocyte aggregation in the presence of mCRP. mCRP induced significant U937 aggregation at 3 h and 24 h ((**E**,**F**), respectively; 100 µg/mL; arrows). Controls (**A**,**B**), nCRP (**C**,**D**) and LPS (**G**,**H**) did not produce clustering of the cells, although clusters were evident even after 3 h of mCRP treatment ((**E**)-arrow). Mean aggregation size of clusters as determined by point to point measurement is shown in (**I**), *p* < 0.0001 **** (note; bars are only seen for the mCRP treated cells since other treatments or control cells did not show any aggregation). Photos were taken in live field at ×100 and 2 random and distinct regions chosen per slide (n = 3) and manually counted for statistical analysis. The experiment was repeated twice—a representative example is shown.

**Figure 2 ijms-25-03097-f002:**
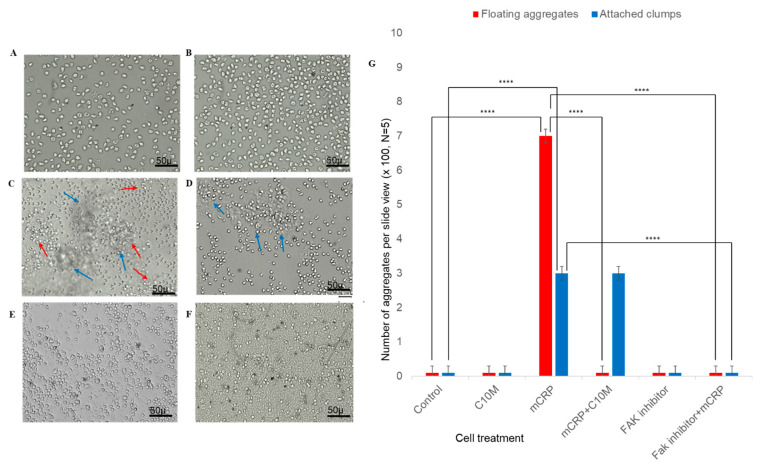
mCRP-induced aggregation of peripheral blood monocytes. (**A**) shows control cultured peripheral monocytes (×100) with no clustering, which have a regular shape and size. (**B**) C10M at 100 µg/mL had no effect on the clustering or appearance of the cells. (**C**) shows the effect of mCRP, (100 µg/mL; 24 h), with appearance of clusters of cells (red arrows, *p* < 0.0001 ****) and adherent clumps (blue arrows, *p* < 0.0001 ****) shown in the representative sample. In (**D**), after C10M pre-incubation for 2 h, small clumps of adherent cells remained (blue arrows), whilst floating aggregates of monocytes disappeared (*p* < 0.0001 ****). (**E**) shows no clustering in the presence of the FAK inhibitor used at 10 µg/mL (24 h), whilst (**F**) (2 h pre-incubation) shows complete inhibition of mCRP-induced clustering and adherence (100 µg/mL, 24 h exposure, *p* < 0.0001 ****). (**G**) shows the mean number of aggregations both floating (red) and attached (blue) from three random areas of each slide and two independent experiments. The experiment was repeated twice—a representative example is shown.

**Figure 3 ijms-25-03097-f003:**
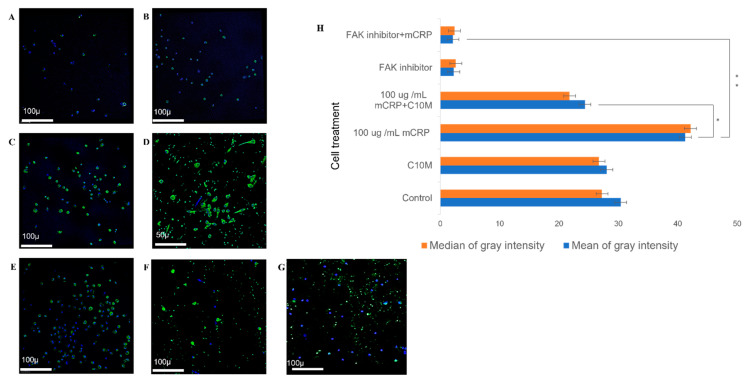
Confocal analysis of FAK expression following mCRP treatment. Peripheral monocytes treated with mCRP (100 µg/mL; 24 h) showed increased expression of p-FAK (increased intensity and number of p-FAK positive cells; (**C**,**D**); ×100), compared with control cells (**A**) or C10M treated (**B**). (**D**) shows a magnified image of mCRP-treated cells (×200), in which a modified morphology suggestive of macrophage differentiation can be seen. Pre-incubation with the CRP inhibitor C10M (100 µg/mL; 24 h) reduced but did not abolish p-FAK expression and nuclear translocation (**E**). Pre-incubation with FAK inhibitor (10 µg/mL) blocked the increased expression of p-FAK induced by mCRP (**G**). (**F**) shows p-FAK expression with inhibitor only. The mean (significance represented by *p* ≤ 0.05 *; and *p* < 0.01 **) and median gray intensity of staining per field of view is shown in (**H**) (three separate fields of view from each of 3 wells; ×100, ImageJ). The experiment was repeated twice—a representative example is shown.

**Figure 4 ijms-25-03097-f004:**
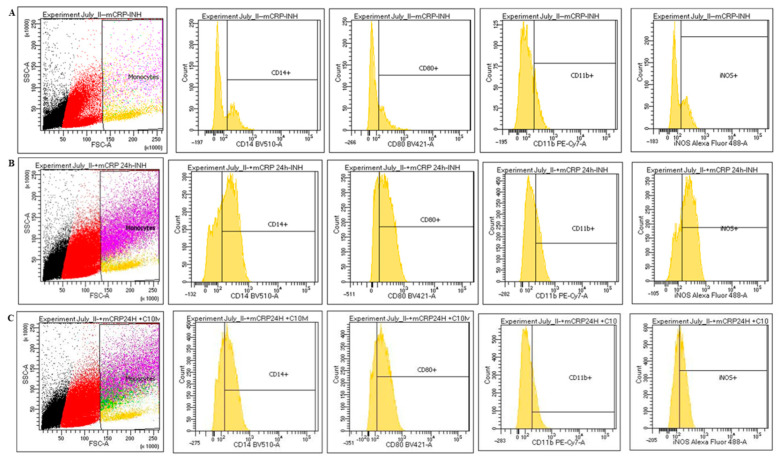
mCRP-modified cell surface markers associated with M1 phenotype, as shown by FACS analysis. Blood monocytes exposed to mCRP (100 µg/mL; 24 h) showed increased cell surface expression of CD11b, CD14 and CD80 (15–27%; 31–63%; and 25–69% respectively), indicating M1 phenotypic shift (**A**,**B**). Pre-incubation with C10M inhibitor (100 µg/mL; 2 h), partially blocked CD80 expression (50%), compared with mCRP alone (**C**). iNOS expression was similarly increased by mCRP (**B**). Pre-incubation of monocytes with the C10M inhibitor also blocked mCRP-induced iNOS by approximately 50%, compared to untreated cells (**C**). Results are shown from a representative experiment of 2, in order to ensure reproducibility of the data.

**Figure 5 ijms-25-03097-f005:**
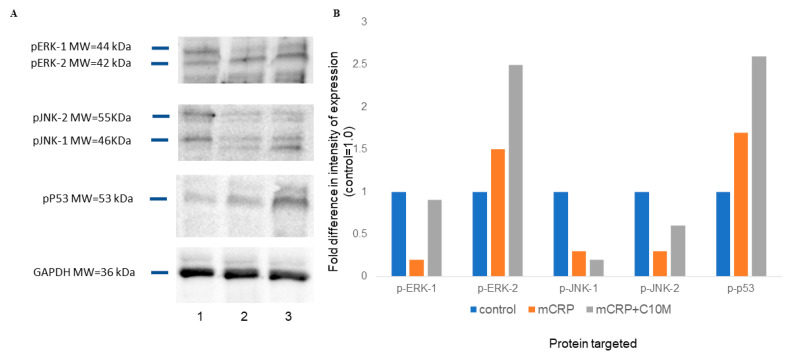
Western blot analysis of mCRP-induced protein phosphorylation. Human peripheral-derived monocytes were differentiated into macrophages (PMA; 50 µg/mL/72 h) cultured in SPM for 48 h and treated with mCRP (100 µg/mL; 8 min), with or without C10M inhibitor 100 µg/mL; 2 h pre-incubation). A reduction in pERK-1, and pJNK-1/2 was seen, compared to the housekeeping control (GAPDH) (**A**). Co-incubation of cells with C10M had no effect. Incubation with mCRP increased the expression of p-p53 (1.7-fold), which was further increased following pre-incubation with C10M (2.6-fold). (**B**) shows semi quantitative data obtained via reading of blot image intensities generated by Image Lab software. The experiment was repeated twice, and a representative example is shown.

**Figure 6 ijms-25-03097-f006:**
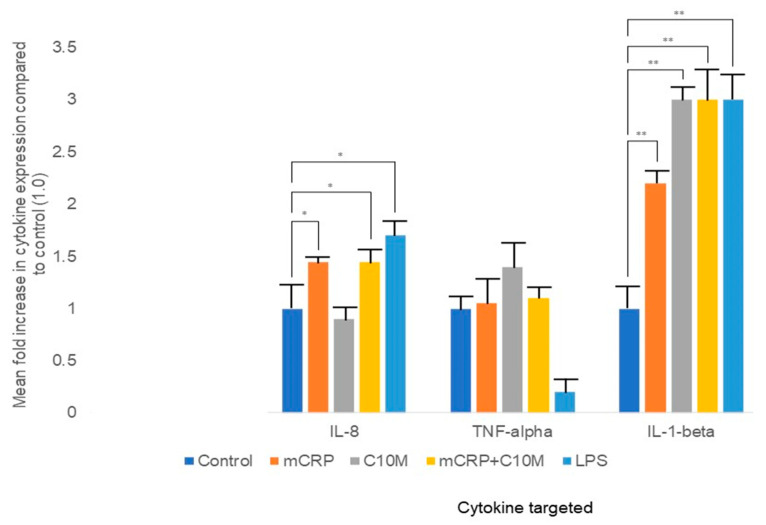
mCRP increased pro-inflammatory cytokine expression in macrophages obtained from peripheral blood monocytes. mCRP (100 µg/mL; 24 h) notably increased the monocyte secretion/expression of IL-8 and IL-1β, but not TNF-α, in the culture medium measured 24 h after addition of mCRP (100 µg/mL), as determined by ELISA (Figure 6). LPS stimulation (positive control) also resulted in an increase in IL-8 and IL-1β expression. Pre-incubation with C10M did not reduce the pro-inflammatory activity of mCRP. Analysis was conducted in triplicate wells, and a representative example of the results from two experiments is shown (significance denoted by * *p* ≤ 0.05, ** *p* ≤ 0.01; ANOVA).

## Data Availability

Data is contained within the article.

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
