# Peer review of "mCRP-Induced Focal Adhesion Kinase-Dependent Monocyte Aggregation and M1 Polarization, Which Was Partially Blocked by the C10M Inhibitor"

_ijms, 2024, doi:10.3390/ijms25063097_

Round 1
Reviewer 1 Report
Comments and Suggestions for Authors
The article is written very poorly, the whole manuscript needs to be checked by a native writer.
The last few lines of the abstract need to be edited.
The font sizes are different in the manuscript at various places, as seen clearly in the discussion.
The statement made by the author in the first line of the introduction is not valid and the author should not claim it.
All figures need to be made clear. The authors mentioned that images are taken on 100x and 200x while it seems like they capture them on 10x and 20x. Provide clear images in every figure.
Bar graphs don't have a Y-axis, while the X-axis is unclear.
FACS data shows variation in gating, while it should be consistent.
Similarly, it is not clear what the author wants to tell in Figure 6, it should be self-reflective.
It is a preliminary study, which needs to be validated in invivo
The author's contribution is not clear, it should tell the work of the author, and what they did.
The funding statement should be written clearly in the English language.
Comments on the Quality of English LanguageVery poor
Author Response
The article is written very poorly, the whole manuscript needs to be checked by a native writer.
We apologise to the reviewer, the whole manuscript has undergone critical editing by a native speaker and we hope now reacges the acceptable level of accuracy.
The last few lines of the abstract need to be edited.
The last part of the abstract has been modified to read:
“In conclusion, mCRP modulates PBMs through a mechanism that involving FAK and results in cell clustering and adgesion concomitant with changes consistant with M1 phenotypical polarization. C10M has potential therapeutic utility in blocking the primary interaction of mCRP with the cells, for example, protecting against monocyte accumulation and residence at damaged vessels that may predispose to plaque development and atherosclerosis.”
The font sizes are different in the manuscript at various places, as seen clearly in the discussion.
We apologise for this-it does not appear in our draft vesion but we have reformatted the whole manuscript and removed this anomoly thank you for pointing this out to us.i
The statement made by the author in the first line of the introduction is not valid and the author should not claim it.-
We have changed the opening line to read “Inflammation is an important damaging”
All figures need to be made clear. The authors mentioned that images are taken on 100x and 200x while it seems like they capture them on 10x and 20x. Provide clear images in every figure.
We thank the reviewer for noticing this-indeed the magnification used in the monocyte images was x 100 (10 x objective)-[although part D of figure 3 was taken at x 200 (20 x objective)] throughout and this has been corrected throughout the text and figure legends. In addition, the figures have been equalised with a maximal improvement in quality-taking into account the specified multiple planes of the cells which results in some dys-focus. The fonts are increasd to increase clarity and all other details scrutinised.
Bar graphs don't have a Y-axis, while the X-axis is unclear.
All the X and Y axis for each figure are now labelled and improved for clarity we apologise for this.For example, In addition, the Median gray intensities are qualified by Image J explanation of analysis in the figure legend for Figure 3.
FACS data shows variation in gating, while it should be consistent.
The FACS gating was designated at the same point for each series (marker) but may be variable between different measured cell surface markers-this is a standard way to show the results we are sorry this was not clear and it has now been added to the methodology in the form of a statement.
Similarly, it is not clear what the author wants to tell in Figure 6, it should be self-reflective.
An additional description has been added to figure 6 legend expaining clearly what the figure shows.
It is a preliminary study, which needs to be validated in in vivo
We agree the study should now be validated in vivo-as the potential importance of aggregation induced by mCRP could be important. We have mentioned this within the discussion section in the last paragraph.
The author's contribution is not clear, it should tell the work of the author, and what they did.
Apologies this has now been clarfied in the text
The funding statement should be written clearly in the English language.
This has been converted to English language-apologies
English language and grammar corrections
The whole manuscript has been carefully assessed and corrected for English language errors
Reviewer 2 Report
Comments and Suggestions for Authors
This manuscript demonstrates that mCRP can influence monocyte aggregation and M1 polarization through the FAK pathway. Novel inhibitors of mCRP, C10M, and FAK inhibitors can block this process. However, the study lacks substantial workload, shows limited innovation, and raises the following concerns.
-
1. My major concern is that some data in this manuscript does not align with previously published results, and the authors have not provided explanations. For example, in Fig5, phosphorylation of JNK does not increase after mCRP stimulation. However, a referenced article [PMID: 34984018] indicates that mCRP activates M1 polarization through JNK phosphorylation, exacerbating disease. The explanation for the decreased activation of the JNK pathway is vague, and I hope for a detailed explanation, highlighting differences with the cited literature. Additionally, the changes in TNF-α levels in Fig6 are perplexing, with a decrease after LPS treatment and minimal increase in the mCRP group, contrasting common knowledge [PMID: 30319609]. Also, the significant rise in IL-1β after adding C10M is not discussed in the manuscript. Similarly, in Fig3, while the C10M group and mCRP+C10M group may not have statistical significance, the lower p-JNK levels after adding mCRP are surprising.
-
2. The second concern is the poor quality of the images in the article. None of the representative images are labeled with group information, making it challenging to interpret them. In Fig1E, I cannot see the indicated arrows, and the scale in Fig1 is unclear. The distinction between blue fluorescence and green fluorescence
-
in Fig3 is not explained. Fig3D, supposedly a zoom-in of Fig3C, appears to be from a different region. The brightness in Fig3F/G seems inconsistent with other images, impacting the reliability of statistical results. Overall, low-quality images significantly affect the reliability of the manuscript and disrupt the reader's comprehension.
-
3. The third concern is that the manuscript merely describes the apparent activation of JNK phosphorylation and nuclear translocation by mCRP, leading to monocyte aggregation and M1 polarization. Other studies have reported M1 polarization and increased inflammatory factors due to mCRP, as well as the activation of the JNK pathway. Given the known FAK-JNK pathway, the overall contribution of this manuscript appears limited. The manuscript attempted to exclude the role of nCRP but does not delve into the underlying mechanisms.
-
4. On manuscript Page 6, there is a passage that begins with "M for 2h…". It is unclear whether this passage is part of the main text or a figure legend.
-
5. In section 3.3 of the main text, it is mentioned that "LPS stimulation (positive control) also resulted in an increase in MCP-1 expression," but there seems to be no result for the MCP-1 indicator in Fig6.
-
6. The article appears to use peripheral blood from 2 human donors, but there is no mention of relevant ethical approval.
The primary issue with the English in this manuscript lies in the unclear and somewhat vague references in the figure legend sections, creating obstacles to smooth reading.
Author Response
This manuscript demonstrates that mCRP can influence monocyte aggregation and M1 polarization through the FAK pathway. Novel inhibitors of mCRP, C10M, and FAK inhibitors can block this process. However, the study lacks substantial workload, shows limited innovation, and raises the following concerns.
- My major concern is that some data in this manuscript does not align with previously published results, and the authors have not provided explanations. For example, in Fig5, phosphorylation of JNK does not increase after mCRP stimulation. However, a referenced article [PMID: 34984018] indicates that mCRP activates M1 polarization through JNK phosphorylation, exacerbating disease.
- The explanation for the decreased activation of the JNK pathway is vague, and I hope for a detailed explanation, highlighting differences with the cited literature. Additionally, the changes in TNF-α levels in Fig6 are perplexing, with a decrease after LPS treatment and minimal increase in the mCRP group, contrasting common knowledge [PMID: 30319609].
Response to 1,2:
Regarding previously published results and our disagreement. It is indeed interesting –we have previously used exactly the same technique to show p-ERK for example increased dramatically after 8 minutes exposure to mCRP in endothelial cells see- Slevin M et al (2015)-Sci Rep- Monomeric C-reactive protein-a key molecule driving development of Alzheimer’s disease associated with brain ischaemia? So, we were surprised by some of these findings with regard to protein phosphorylation. Nevertheless, we have to show our findings, and we are able to back them up with some literature-which we have clarified in the discussion text in more detail- regarding p-ERK1/2 and p-JNK and macrophages:
P-ERK1/2- p-ERK-1 was in fact reduced in the presence of mCRP in our results. He et al (2022 in cell reports) as we have discussed showed that a short term activation of MEK-ERK pathways induced macrophage polarization to M2 (anti-inflammatory)-hene a reduction in ERK-1 phosphorylation as we showed agrees with the hypothesis of maintaining a pro-inflammatory M1 phenotype with no polarization.-M2 polarization was shown within page 7 of this manuscript to be blocked by RAF-MAP kinase inhibitors.- we have extended our discussion to reflect this.There is also data indicating later shifts in ERK phosphorylation might be relevant for inflammatory action and this was a limitation of our study-added also to the Discussion.
P-JNK1/2 Zha et al used the leukaemia cell line THP-1 (likely mutated) monocytes stimulated chronically by mCRP for 24h.In addition the images are not convincingly different in the in vitro section for increased phosphorylation of p-JNK isoforms. No work to our knowledge has used and investigated acute changes in JNK with primary peripheral blood obtained differentiated monocytes. –an explanation has now been included in our results section showing the differences between the models and methodology which could explain the results..
TNF-α—Jundi et al in 2020 showed that neither TNF-alpha nor Il-6 secretion was inceased in U937 cells exposed to mCRP FOR 24h, whilst Krayem et al (2017), showed a reduction in expression of TNF-alpha and no change in IL-1-beta under the same experimental conditions-these are the main papers exploring this phenomenan and hence our results match this –for mCRP, as we now have added to the discussion. Re the LPS, this is difficult to explain considering that IL-1 beta levels were raised substantially by the same LPS in the same samples. We have added a comment at least to indicate our notation of this result (also we used a lower 10ng/ml than is sometimes used in other studies and have noted this also in the text)
Also, the significant rise in IL-1β after adding C10M is not discussed in the manuscript. Similarly, in Fig3, while the C10M group and mCRP+C10M group may not have statistical significance, the lower p-JNK levels after adding mCRP are surprising.
IL-1-beta was increased in the presence of C10M inhibitor only. This would be the first time it has been examined within this monocyte-macrophage system. Zeller et al (2022), used an indirect activation – ADP-stimulated platelets- to show that C10M reduced IL-1 beta expression, measured not by ELISA but flow cytometry and confocal microscopy, I think this is quite a different methodology, however, increased IL-1beta in the presence of C10M alone deserves an explanation as the reviewer rightly identifies. Somehow could the binding of C10M to the membrane of the monocytes, stimulate release of the active form of Il-1- beta from the cytoplasmic store? We can only hypothesise but we have added to the text this suggestion and explained as a limitation that this require further detailed investigation.
- The second concern is the poor quality of the images in the article. None of the representative images are labeled with group information, making it challenging to interpret them. In Fig1E, I cannot see the indicated arrows, and the scale in Fig1 is unclear. The distinction between blue fluorescence and green fluorescence in Fig3 is not explained. Fig3D, supposedly a zoom-in of Fig3C, appears to be from a different region. The brightness in Fig3F/G seems inconsistent with other images, impacting the reliability of statistical results. Overall, low-quality images significantly affect the reliability of the manuscript and disrupt the reader's comprehension.
Regarding the images: Labels have been applied to each axis. In Fig 1E, the arrows have been converted to blue and they can now be easily visualised.The scale bars have been improved to be seen we apologise for this. Figure 3D is not meant to be a magnified version of 3C we apologise for this misunderstanding and we have clarified in the figure and text. The brightness of all subparts of Figure 3 has been equalised (nuclear staining is now also clearly seen) and the statistics remeasured although the final intensities are almost identical to the original calculated gray means.. All the images have been optimised.
4 The third concern is that the manuscript merely describes the apparent activation of JNK phosphorylation and nuclear translocation by mCRP, leading to monocyte aggregation and M1 polarization. Other studies have reported M1 polarization and increased inflammatory factors due to mCRP, as well as the activation of the JNK pathway. Given the known FAK-JNK pathway, the overall contribution of this manuscript appears limited. The manuscript attempted to exclude the role of nCRP but does not delve into the underlying mechanisms.
Novelty and limitations of the work:The novel elements of the work are that we are first to describe the ability iof mCRP to directly induce monocyte aggregation and increase adherence concomitant with macrophage phenotypical changes towards pro-inflammatory M1 through a FAK-dependent mechanism. Specific to your comments:
-It was FAK nuclear translocation that we showed not JNK and FAK is one of the major novel findings of mCRP signalling not studied before. However, only by Zha et al (2021)was mCRP-induced M1 polarization shown but they used the oncogenic THP-1 cells (limitations) and did not identify the novel aggregation and FAK-associated signalling as we did nor investigate the effects of the novel inhibitor C10M-we have added to the discussion highlighting these points.
5 On manuscript Page 6, there is a passage that begins with "M for 2h…". It is unclear whether this passage is part of the main text or a figure legend.
Correction: apologies we have corrected this
- In section 3.3 of the main text, it is mentioned that "LPS stimulation (positive control) also resulted in an increase in MCP-1 expression," but there seems to be no result for the MCP-1 indicator in Fig6.
Correction-we have removed this-it was part of other work not included in the manuscript
- The article appears to use peripheral blood from 2 human donors, but there is no mention of relevant ethical approval.
Addition-ethics approval has been added to the methodology section.
- English language revision
The whole manuscript has been carefully edited for English language mistakes.
Round 2
Reviewer 2 Report
Comments and Suggestions for Authors
Thank authors for their explanation of all my questions. The add part in Discussion makes the manuscript more logical. But the revision of the manuscript seems to have significant layout issues. Figure legends are mixed with the main text, and all subheading numbers are problematic. Moreover, phrases like "Western blotting results from adherent 'macrophages'" cannot be used as subheadings as they fail to highlight the scientific content that follows. The current version of the manuscript is non-readable. Please submit a correctly formatted manuscript.
One more thing is that the background of Fig 3A/B is too high. Please adjust it.
Author Response
R-Thank authors for their explanation of all my questions. The add part in Discussion makes the manuscript more logical. But the revision of the manuscript seems to have significant layout issues.
Figure legends are mixed with the main text, and all subheading numbers are problematic. Moreover, phrases like "Western blotting results from adherent 'macrophages'" cannot be used as subheadings as they fail to highlight the scientific content that follows. The current version of the manuscript is non-readable. Please submit a correctly formatted manuscript.
One more thing is that the background of Fig 3A/B is too high. Please adjust it.
A-Thank you for the undoubtedly useful comments.
We corrected the layout issues, subheading numbers, and the figure legends are now separated from the main text. Therefore, the manuscript is now correctly formatted. We apologize for the inconvenience, as we submitted the manuscript using a regular word document and were not aware of the changes made when the IJMS template was implemented by the Journal.
"Western blotting results from adherent 'macrophages'" changed to "3.2. Western blotting showed that mCRP reduced MAP-kinase phosphorylation but increased p53 after 8 minutes exposure".
The background of Fig.3A/B has been adjusted.
We would like to genuinely thank the Reviewer for their inputs!
Round 3
Reviewer 2 Report
Comments and Suggestions for Authors
Thank authors for revision. I have no more major question. The only thing I want to point out is that on Methods part and subtitle of 3.2, it was described the time of mCRP stimulation as 8 minutes, but in the main text and Figure 5 legend, the mCRP treatment is 10 minutes. Please confirm which one is the actual condition for your experiment.
Author Response
Many thanks for the comment we have corrected the timing of the mCRP to 8 minutes in both